# The Evaluation of Lipid-Lowering Treatment in Patients with Acute Coronary Syndrome in a Hungarian Invasive Centre in 2015, 2017, and during the COVID-19 Pandemic—The Comparison of the Achieved LDL-Cholesterol Values Calculated with Friedewald and Martin–Hopkins Methods

**DOI:** 10.3390/jcm13123398

**Published:** 2024-06-11

**Authors:** Laszlo Mark, Péter Fülöp, Hajnalka Lőrincz, Győző Dani, Krisztina Fazekas Tajtiné, Attila Thury, György Paragh

**Affiliations:** 1Cardiology Department, Bekes County Central Hospital Pandy Kalman Branch, 5700 Gyula, Hungary; 2Division of Metabolism, Department of Internal Medicine, Faculty of Medicine, University of Debrecen, 4032 Debrecen, Hungary; 3Laszlo Elek Town Hospital, 5900 Orosháza, Hungary; 4Central Laboratory, Bekes County Central Hospital Pandy Kalman Branch, 5700 Gyula, Hungary

**Keywords:** acute coronary syndrome, COVID-19 pandemic, LDL-cholesterol, Friedewald formula, Martin–Hopkins method

## Abstract

**Background/Objectives:** Patients with acute coronary syndrome (ACS) represent a vulnerable population. We aimed to investigate serum lipid levels of patients with ACS upon admission and during one year of the COVID-19 pandemic in a rural county hospital, and compared these findings with the data of patients with ACS in 2015 and 2017. The secondary aim of this paper was the comparison of the LDL-C values calculated with the Friedewald and Martin–Hopkins methods. **Methods:** A retrospective analysis of lipid-lowering data of patients treated with ACS in 2015, 2017 and in a COVID-19 year (1 April 2020–31 March 2021) was performed; the patient’s numbers were 454, 513 and 531, respectively. **Results:** In the COVID-19 period one year after the index event, only 42% of the patients had lipid values available, while these ratios were 54% and 73% in 2017 and in 2015, respectively. Using the Friedewald formula, in the COVID-19 era the median of LDL cholesterol (LDL-F) was 1.64 (1.09–2.30) mmol/L at six months and 1.60 (1.19–2.27) mmol/L at one year, respectively. These values were 1.92 (1.33–2.27) mmol/L and 1.73 (1.36–2.43) mmol/L using the Martin–Hopkins method (LDL-MH). The LDL-F yielded significantly lower values (15% lower at six months, *p* = 0.044; and 8% lower at one year, *p* = 0.014). The LDL-F reached the previous target of 1.8 mmol/L during the COVID-19 pandemic 36% at one year vs. 48% in 2017, and 37% in 2015. The recent target LDL-C level of 1.4 mmol/L was achieved in 22% of cases in the COVID-19 pandemic, 16% in 2015 and 19% in 2017. **Conclusions:** A significantly lower proportion of patients with ACS had available lipid tests during the COVID-19 pandemic. Besides the lower number of available samples, the proportion of achieved 1.4 mmol/L LDL-C target lipids was stable. More rigorous outpatient care in the follow-up period may help to improve the quality of lipid lowering treatments and subsequent secondary cardiovascular prevention. If direct LDL-C determination is not available, we prefer the LDL calculation with the Martin–Hopkins method.

## 1. Introduction

Results from previous studies have shown that reducing serum low-density lipoprotein cholesterol (LDL-C) levels inhibits the progression of atherosclerosis [1,2,3,4,5]. Indeed, by lowering LDL-C by more than 50% or reaching less than 2 mmol/L of LDL-C, not only can the progression of atherosclerosis be mitigated, but it may be regressed [6,7]. Consequently, the European Society of Cardiology/European Society of Atherosclerosis (ESC/EAS) recommendation was modified in 2019 aiming to achieve <1.4 mmol/L and at least 50% baseline LDL-C reduction in the very high-risk category; and for those with recurrent acute coronary syndrome within two years, an even stricter target LDL-C level of <1 mmol/L is suggested [8]. Thus, it is not surprising that clinicians should follow the principle of “the higher the risk, the higher the benefit” to improve the quality of life and extend the life expectancy of patients with acute coronary syndrome (ACS).

Despite being an object of lots of negative preconceptions, a lipid-lowering treatment is one of the most important elements of cardiovascular prevention. A high LDL-C level is not only a risk factor for cardiovascular disease (CVD), but it is the main cause of atherosclerosis, too. In addition to the principles of LDL-C lowering (“as low as possible, as early as possible, as long as possible”), it is also essential to emphasize that the greater the cardiovascular (CV) risk, the greater the benefit from the treatment [8].

Assessing the quality of lipid-lowering treatment in 18 countries, the DaVinci study found that the actual recommendation target of 1.4 mmol/L LDL-C was achieved in less than one-fifth of the very high-risk patients, while 41% of the patients reached the former recommendation target of 1.8 mmol/L [9]. The picture is even worse in the central and eastern European countries, with an average of 10% reduction in the likelihood of reaching this target [10]. According to the Hungarian Cardiology Position Statement, inadequate care, including failure to reduce target lipid levels may play a significant role in unfavourable outcomes. In addition, data from the Hungarian Myocardial Infarction Registry (HUMIR) suggest that the rate of statin recommendation at the hospital discharge of patients with ACS is over 90% [11]. National data about the subsequent period are lacking, but some regional observations are available including our previous reports [12,13].

The COVID-19 pandemic represented an enormous healthcare burden with a whole new set of challenges, with significant changes in doctor–patient encounters. Poor persistence and adherence to lipid-lowering drugs are unlikely to be improved during this period. On the other hand, several retrospective investigations demonstrated that statins, via their anti-inflammatory and antithrombotic effects, had a beneficial impact on disease course and mortality in patients infected with COVID-19 [14,15,16]. In addition to the observational studies, the potential beneficial effect of statins was supported by a Mendelian randomization trial, indicating an increased expression of HMG-coenzyme A reductase and increased risk of infection requiring hospitalization [17]; however, the actual effect of statins on COVID-19 infections has not yet been demonstrated in a randomized prospective study.

Lipid-lowering treatment is known to have pleiotropic effects, the main feature of which is the reduction in inflammation, which also prevailed in the case of COVID-19 infections. The most accessible marker of the anti-inflammatory effect is the improvement of hsCRP. The use of statins has the most proven pleiotropic effects, among others the beneficial impact on platelet function [18,19,20].

During the pandemic, the treatment of ACS remained the priority of cardiology care. Cannata et al. reported a decrease in the number of ACS cases and increased cardiovascular mortality in these patients [21]. A lower ACS rate was also observed in the Hungarian capital, Budapest [22]. These somewhat controversial data indicate alterations in the care of patients with ACS at the time of the event and in the post-event period, as well.

Since in Hungary, the direct LDL-C measurement is not widely available, the calculation of LDL-C is most often based on Friedewald’s formula if the triglyceride (TG) level is less than 4.5 mmol/L [23]. However, the applicability of LDL-C was calculated with Friedewald’s formula (LDL-F), especially for patients with TG levels of almost 4.0 mmol/L and for those who have LDL-C levels under 1.8 mmol/L. Therefore, a decade ago, a more exact, novel method for estimating LDL-C, termed Martin–Hopkins calculation (LDL-HM), has been developed [24]. 

In the retrospective analysis of the Further Cardiovascular Outcomes Research With PCSK9 Inhibition in Patients With Elevated Risk (FOURIER) trial data, it has been shown that the Martin–Hopkins method performs better regarding the undertreatment because of LDL-C underestimation by the Friedewald method [25].

In the present paper, we analysed the changes in lipid levels of patients treated for ACS during the first year of the COVID-19 pandemic in our invasive centre. We investigated lipid levels at the time of the index event (upon admission), 6 months and one year after, with a significant part of the follow-up period falling into the pandemic period, and these results were compared to those of patients who suffered with ACS in 2015 and 2017. The secondary aim of this paper, since direct LDL-C determination is not available in our hospital, was the comparison of the LDL-C values calculated with Friedewald and Martin–Hopkins methods.

## 2. Materials and Methods

### 2.1. Patients

We retrospectively collected data from patients who were hospitalized for ACS and underwent primary coronary intervention between 1 April 2020 and 31 March 2021 (COVID-19 year, baseline index event) at the Invasive Division of Cardiology Department of the Bekes County Central Hospital Pandy Kalman Branch, Gyula, Hungary. We also collected patients’ data after 6 and 12-month periods after discharge. Data collection was closed on 31 December 2022. We compared their data with patients with ACS who were hospitalized in the year 2017 (from 1 January 2017 to 31 December 2017) and 2015 (from 1 January 2015 to 31 December 2015) at the same cardiology department. Patient information was collected from the local and national inpatient and outpatient databases. ACS was diagnosed according to the recent guidelines of the European Society of Cardiology [26]. The main characteristics of related comorbidities (i.e., the proportion of hypertension, diabetes mellitus, hyperlipidaemia, etc.; incidence of statin-related myopathy and use of dietary supplements (omega-3)) were accessed. None of the patients with ACS received omega-3 supplementation. The study was provided according to the Helsinki Declaration. The Institutional Research Ethics Committee of the Bekes County Central Hospital granted permission to conduct the study (Approval code: 263/2016; Approval date: 12 December 2016).

### 2.2. Determination of Routine Laboratory Parameters

After 12 h of fasting, 10 mL of venous blood was taken from the patients. Serum samples were separated by centrifugation at 3500× *g* for 10 min at 4 °C. From the freshly taken samples, lipid parameters were determined according to the standard laboratory methods at the Central Laboratory Bekes County Central Hospital Pandy Kalman Branch, Gyula, Hungary using a Cobas 600 analyser (Roche Ltd., Basel, Switzerland). Serum total cholesterol and TG concentrations were determined using an enzymatic, colorimetric method; in the case of HDL-C, a homogeneous, enzymatic method was used (Roche HDL-C plus 3rd generation). The tests were carried out according to the instructions given by the manufacturer. Non-HDL-C was calculated by subtracting HDL-C from total cholesterol.

### 2.3. Calculation of LDL-C

LDL-C was calculated by the Friedewald formula (LDL-F) as LDL-C = total cholesterol (TC)—high-density lipoprotein cholesterol (HDL-C)—triglycerides (TG)/2.2), if the TG level is less than 4.5 mmol/L, since direct LDL-C determination is not routinely performed in our hospital. To analyse the level of patient care, we compared these levels with those who were treated for the same diagnosis in our department in 2015 and 2017. As TG levels above 2.3 mmol/L may significantly interfere with LDL-F [27], we also used the Martin–Hopkins method (LDL-MH) [24] at the index event and 6 and 12 months after to compare them with LDL-F. The Friedewald equation assumes a fixed factor of 5 for mg/dL or 2.2 for mmol/L for the ratio of TG to very low-density lipoprotein cholesterol (TG/VLDL-C); however, the actual TG/VLDL-C ratio varies significantly across the range of TG and cholesterol levels. Using the LDL-MH, the LDL-C estimates were derived as the (non-HDL-C) triglycerides/adjustable factor, where the adjustable factor was determined as the strata-specific median TG/VLDL-C ratio.

### 2.4. Statistical Methods

Statistical analyses were performed using the SPSS 23.0 for Windows 10^®^ (SPSS, Chicago, IL, USA) software package. Kolmogorov–Smirnov and Shapiro–Wilk tests were used for testing the normality of the data distribution. Categorical data were presented in absolute numbers and percentages. Continuous variables were presented with arithmetic mean and standard deviation for normal distributions, and median and interquartile ranges for non-normal distributions. Categorical data were analysed using Fisher’s exact test, while comparisons of continuous variables were made using a non-parametric Mann–Whitney U test. All *p* values shown are two-sided where *p* < 0.05 was considered to be significant.

## 3. Results

During the COVID-19-year follow-up period, 607 patients were hospitalized with ACS, but 49 men and 27 women (9.2% and 5.1%, respectively) were deceased. Their lipid values were not included in our analysis. Therefore, a total of 531 patients—including 312 males and 219 females (58.8% vs. 41.2%), with a mean age of 66.9 ± 12.2 years for men and 72.4 ± 11.2 years for women—were analysed. The study design flowchart of enrolled subjects is depicted in Appendix A. Additionally, detailed characteristics of metabolic profiles in patients with ACS, including the proportion of ST-elevation myocardial infarction (STEMI), non-ST-elevation myocardial infarction (NSTEMI), unstable angina pectoris (UAP), peripheral artery disease (PAD), etc., as well as the medical history of hypertension, diabetes mellitus, hyperlipidaemia and smoking, are summarized in Table 1.

Total cholesterol concentrations, LDL-C, calculated by the Friedewald formula, as well as levels of HDL-C, non-HDL-C and TG are presented in Appendix A. Compared with the baseline results, total cholesterol levels decreased by 27% at 6 months after the index event (*p* < 0.001) and by 28% at 12 months (*p* < 0.001). LDL-C levels decreased by 49% and 50% (*p* < 0.001 and *p* < 0.001, respectively), while non-HDL-C levels showed a 37% reduction at both time points (*p* < 0.001 and *p* < 0.001, respectively). On the other hand, HDL-C and TG levels were significantly unchanged.

The lower part of Table 1 indicates the distribution of the lipid-lowering therapy at discharge in 2015, 2017 and the COVID-19 year. Having no information on the reason, 3% of the patients did not receive lipid-lowering treatment at discharge during the COVID-19 pandemic, while these ratios were 1% in 2017 and 7% in 2015, respectively. On the other hand, there were no patients on low- and moderate-intensity statin in the COVID-19 year and 2017, whereas 1% of the patients received this regime in 2015. In respect of the high-intensity therapy, 88% of patients with ACS were advised to receive this regime, while these ratios were 96% and 86% in 2017 and 2015, respectively. A significantly higher portion of patients with ACS received statin + ezetimibe combination therapy during the COVID-19 year than in 2015 and 2017 (*p* < 0.001). None of the patients with ACS received the PCSK9 inhibitor at discharge and during the follow-up period. Additionally, we did not detect side effects (e.g., myopathy) about lipid-lowering treatment during the whole follow-up period.

Table 2 compares the median LDL-C levels at the index event in the COVID-19 year, then 6 and 12 months after, using the LDL-F and the LDL-MH method. At the index event, LDL-F was 3.20 (2.30–4.19) mmol/L, which changed to 1.64 (1.09–2.30) mmol/L after 6 months and 1.6 (1.19–2.27) mmol/L after 12 months, respectively. LDL-MH turned out to be 3.32 (2.35–4.27) mmol/L at the index event, 1.92 (1.33–2.27) mmol/L after 6 months and 1.73 mmol/L after 12 months, respectively. There was no significant difference between the results of the two calculations at the index event. After 6 months, LDL-MH indicated a 15% higher level compared with LDL-F (*p* = 0.044) and remained 8% higher after 12 months (*p* = 0.014).

During the investigated pandemic year, we found laboratory data on lipid concentrations 6 months after discharge in only 35% of the patients, while this ratio was 43% after 12 months. Of note, these ratios were 54% and 53% in 2017 and 55% and 73% in 2015, respectively (Table 1). Unfortunately, no laboratory data were found for the remaining patients. 

Using the two calculations, the frequency of achieving the previous LDL-C target of 1.8 mmol/L and the recent 1.4 mmol/L is also shown in Table 2**.** After 6 months, the previous (1.8 mmol/L) LDL-C goal was achieved in 32% and 28% of the patients using the LDL-F and the LDL-MH calculations, respectively. At the same time point, the recent and more rigorous target was attained in 22% and 19% of the patients with LDL-F and LDL-MH, respectively. After 12 months, the 1.8 mmol/L LDL-C goal attainment was significantly higher (36% by LDL-F and 27% by LDL-MH; *p* < 0.001), while 1.4 mmol/L LDL-C goal attainment rates were found to be 22% and 26% with LDL-F and LDL-MH, respectively.

Figure 1 shows the LDL-C target attainment rates based on the LDL-F and LDL-MH calculations in our patients post-ACS in 2015, 2017 and the COVID-19 period. As mentioned above, 36% of our patients achieved the previous target of 1.8 mmol/L LDL-C levels during the COVID-19 pandemic, while we were able the reach the same goals of 48% in 2017 and 37% in 2015. When analysing the 1.4 mmol/L LDL-C goal attainment rates, 19% of the patients reached this target after 12 months of treatment in 2017, compared to 22% during the early period of the COVID-19 pandemic.

## 4. Discussion

Patients with ACS are one of the most vulnerable groups of patients in cardiology practice where guideline-based treatment is of paramount importance following the principle of “the greater the risk, the greater the benefit” [28,29]. Leaving behind the previous target levels, the 2019 ESC/EAS recommendation calls for a 1.4 mmol/L LDL-C goal in patients with a very high CV risk, even pinning it potentially down to 1.0 mmol/L for patients with recurrent vascular events within two years [8]. Data from the Swedish Infarction Registry (SWEDEHEART) show that the incidence of major cardiovascular events, all-cause mortality and recurrent myocardial infarctions decreased proportionally to the degree of the achieved LDL-C reduction [30]. On the other hand, recurrent myocardial infarction and all-cause mortality are inversely related to statin adherence [31]. Indeed, a decreased incidence of cardiovascular events was observed in parallel with an increased intensity of lipid-lowering treatment and increased adherence; however, a gradually attenuating treatment intensity was experienced over time [32]. 

During the COVID-19 period, a decreased number of ACS cases was described in our country [21,22], which was mainly due to the reduced number of non-ST-elevation myocardial infarctions [21,33]. Oppositely, we did not experience a decrease in the ACS cases as we treated 531 patients with ACS in the study period while the case numbers were 513 and 454 in 2017 and 2015, respectively. 

Despite the obvious difficulties of providing up-to-date patient care during COVID-19, we were able to reach significant reductions in several major lipid parameters including total cholesterol, LDL-C and non-HDL-C. Regarding HDL-C and TG levels, the changes were not significant; however, one might speculate that proper maintenance of lipid-lowering treatment and patient adherence would result in a long-term cardiovascular risk reduction and the decreased reoccurrence of cardiovascular events in our patients. Further studies and a longer investigating time frame, including the other COVID-19 waves, could answer these speculations. 

It is important to note, that in most of the cases, physicians also have to initiate high-intensity statins to provide effective lipid control. Having no data on the potential contraindications or other drawbacks, we observed an increased number of omitting statins upon discharge with a decreased number of initiating high-intensity statins for these very high-risk patients. On the other hand, an increased rate of statin + ezetimibe combination therapy was observed. Ezetimibe is recommended as a first-line treatment following ACS as recommended by the expert statement [34]; however, local health insurance regulations may hamper the adoption of these guidelines. Nevertheless, a greater awareness of cardiovascular risk reduction and secondary prevention is needed both from the cardiologists and the decision-makers, as well.

In line with previous data, we calculated significantly higher LDL-C levels with the Martin–Hopkins method compared to the Friedewald formula [35]. LDL-MH is considered to be more accurate, especially in conditions with elevated TG levels [36]; however, direct LDL-C measurements should be prioritized nationwide including in rural healthcare providers. The achievement rate of LDL-C targets is an important characteristic factor regarding the quality of lipid-lowering therapy. The differences observed in the LDL-C calculations are also reflected in the target attainment rates. Regardless of the calculations used, the LDL-C goal attainment rate was inherently low and further highlights the necessity of aggressive lipid-lowering therapy.

We also compared target attainment efficacy during the COVID-19 pandemic with earlier results in our centre. Regarding the previously suggested target levels, slightly more than one-third of the patients reached their target values during the pandemic, which was comparable to the attainment rate observed in 2015. Even with the peak in 2017, only about half of the patients were on their LDL-C goal. Considering the recent target levels, about one-fifth of patients reached the LDL-C targets, with a slightly better tendency in time. In the past, the major problem of patient follow-up was that a significant proportion of patients post-ACS did not show up on control appointments, which worsened during the COVID-19 pandemic. Its existence, however, does not completely explain the dramatic decrease in the number of patients attending follow-up appointments. Indeed, physicians do need to pay more attention to act in line with the 2021 Hungarian expert position paper promoting the provision of laboratory referral and already fixed follow-up appointments at the discharge of patients with ACS with a greater involvement of primary care providers [37].

The poor target LDL-C attainment rate is not only characteristic of our hospital; it can be said to be a worldwide phenomenon. Figure 2 shows the proportion of patients at a high risk of reaching the LDL targets of 1.8 mmol/L and 1.4 mmol/L based on Hungarian and European data [9,38,39]. In the last columns, after the publication of the 2019 ESC lipid guidelines, the percentage of patients reaching the target of 1.4 mmol/L is indicated as well. In our study, 48% of patients achieved the LDL target of 1.8 mmol/L in 2017, while only 36% achieved it in 2020–2021. This trend suggests that LDL-C reduction, which is crucial for cardiovascular prevention in patients, has lagged behind in the COVID-19 period compared to previous years.

Besides the insufficient target attainment rate, another problem is that after one year, only 53% of patients post-ACS had lipid results in 2017 (although the 1.8 mmol/L LDL-C target has been reached by 48% of patients, which, as shown in the figure, is quite a good result, better than that of DaVinci). One year after the ACS in the COVID-19 period, 43% of patients had a lipid result. In 2015, 73% of the patients post-ACS had lipid results in the database. These data show that, compared to previous years, a smaller percentage of patients had a lipid test in the COVID-19 period. Overall, it can be stated that the control of these very vulnerable patients is insufficient.

In addition to the statins, ezetimibe administration should also play a more important role in lipid-lowering and subsequent CV risk reduction. According to a recent position paper, ezetimibe should also be included in the first-line treatment of patients with ACS at discharge [37]; however, national regulations limit the first-line use of ezetimibe for those with recurrent cardiovascular events already receiving statin therapy. HUMIR data indicate that about 25% of patients with ACS undergoing coronary intervention have already endured some vascular event previously. 

Even during the COVID-19 year the great majority of patients with ACS were discharged with statins (only 3% of the patients left the hospital without this recommendation (Table 1). In a previous national analysis, it was established that the reason for the insufficient lipid reduction is not in the statin recommendations of patients discharged from invasive centres, but in the absence or inadequacy of controls following the event [11]. It is also gratifying that, despite the difficulties caused by the COVID-19 pandemic, the proportion of patients taking ezetimibe increased (Table 1); the 9% is close to the results of the DaVinci study. 

In-hospital mortality of Hungarian patients treated with ACS is similar to that of those treated in western and northern Europe, with a significant gap in one-year mortality. Indeed, Hungarian post-NSTEMI mortality shows a significant fallback to the same in Sweden (24.6% vs. 14.8%) [40,41]. Organized follow-up care, patient and medical staff education, and thus improved populational health awareness are of major importance in our country. An increased availability of cardiac rehabilitation would also improve mortality, as well [42].

The main limitation of our study is the nature of the single centre, and the number of patients is limited. Additionally, a significant disadvantage is the unavailability of the direct LDL-C values in our hospital; therefore, if direct LDL-C determination is not available, we prefer the LDL calculation with the more precise MH-LDL-C. Due to the retrospective nature of the investigation, any causal relationships should be approached with caution. On the other hand, our patients represent a relatively homogenous population in terms of ACS care and our data could serve as a basis for other comparative investigations. By comparing the results of three separate years for patients with the same diagnosis in the same area, trends can be established, which also show directions for improving lipid-lowering therapy.

## 5. Conclusions

Several conclusions could be drawn up regarding the present study. Firstly, during the COVID-19 period, we did not have fewer patients with ACS, but the rate of achieving LDL-C target values decreased, and the number of patients who had their LDL-C determined within a year after the index event decreased. The rate of statin recommendations for patients discharged due to ACS is adequate, and we must pay attention to increasing the number of statin + ezetimibe combination recommendations taking into account the insurance regulations.

Secondly, strict adherence to the recent lipid-lowering guidelines improves post-ACS mortality. Although statins are generally prescribed to these patients at discharge, regular lipid monitoring and persistence need serious improvement. Focusing on the importance of the above-mentioned, national or regional programs involving primary care providers could be of major help. 

Our study observed a significant difference between LDL-F and LDL-MH in patients with ACS; the latter is closer to a direct LDL-C measurement. In places where a direct LDL-C is not available, it would be worthwhile to use LDL-MH more often, even by a phone application.

Greater attention to the administration of the appropriate dose of statins and ezetimibe, rigorous outpatient care including laboratory referrals and strict follow-up appointments may eventually lead to more effective secondary cardiovascular prevention in ACS.

## Figures and Tables

**Figure 1 jcm-13-03398-f001:**
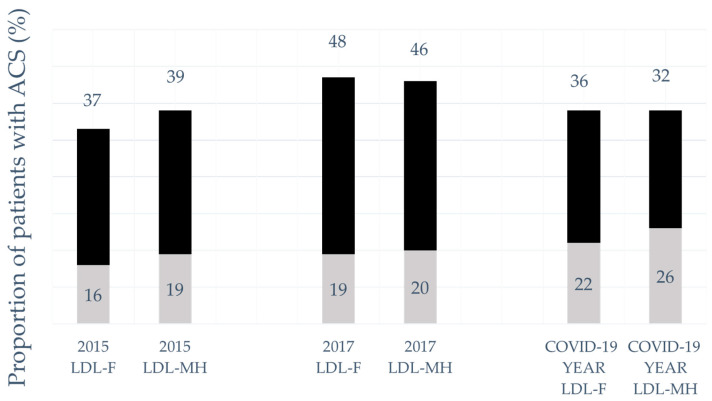
The attainment rate of 1.8 and 1.4 mmol/L LDL-cholesterol target levels based on the calculations with the Friedewald formula (LDL-F) and the Martin–Hopkins method (LDL-MH) in patients with ACS in 2015 (from 1 January 2015 to 31 December 2015), 2017 (from 1 January 2017 to 31 December 2017) and the COVID-19 year (1 April 2020–31 March 2021) in Gyula Hospital. There was no significant difference regarding the LDL-C target attainment in 2015, 2017 and the COVID-19 year (*p* = 0.157). The dark grey part of the bar, with the % inside, shows the 1.8 mmol/L LDL-C attainment rate, and the light grey part of the bar, with the % inside, expresses the 1.4 mmol/L attainment.

**Figure 2 jcm-13-03398-f002:**
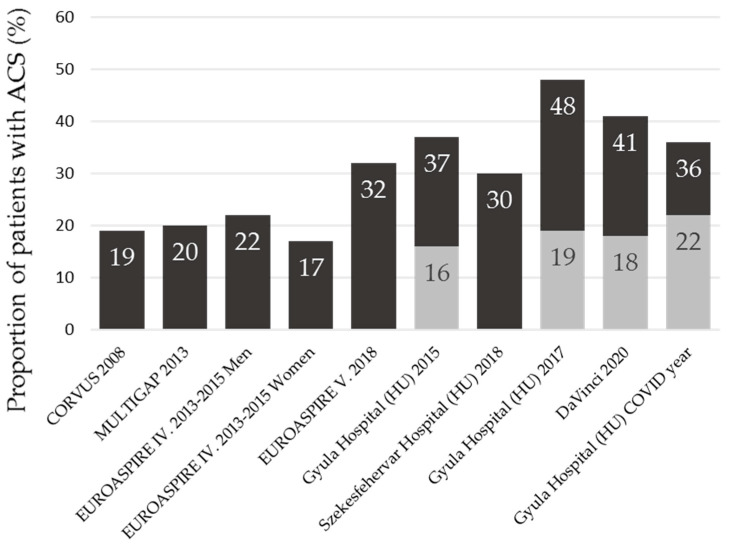
The proportion of patients at a high risk of reaching the LDL targets of 1.8 mmol/L and 1.4 mmol/L based on European and Hungarian data. The dark grey part of the bar, with the % inside, shows the 1.8 mmol/L LDL-C attainment rate, and the light grey part of the bar, with the % inside, expresses the 1.4 mmol/L attainment.

**Table 1 jcm-13-03398-t001:** Characteristics of metabolic profiles in patients with acute coronary syndrome (ACS).

	2015	2017	COVID-19 Year
All patients			
Number of patients	454	513	531
Age (mean ± SD)	68.9 ± 12.4	67.9 ± 12.2	69.2 ± 13.1
Male number (%)	301 (66.3)	306 (59.6)	312 (58.8)
Age (mean ± SD)	66.7 ± 12.0	67.9 ± 12.2	66.9 ± 12.2
Female number (%)	153 (33.7)	207 (40.4)	219 (41.2)
Age (mean ± SD)	73.1 ± 12.1	71.6 ± 12.0	72.4 ± 11.2
ST-elevation myocardial infarction (STEMI)			
STEMI number (% of all ACS)	112 (24.7)	198 (38.6)	271 (51.0)
Age (mean ± SD)	68.6 ± 13.7	66.1± 11.6	65.4 ± 14.4
STEMI male (%)	72 (64.3)	124 (62.6)	154 (63.4)
Age (mean ± SD)	65.8 ± 13.8	64.8 ± 11.9	62.9 ± 11.4
STEMI female (%)	40 (35.7)	74 (37.4)	89 (36.6)
Age (mean ± SD)	73.4 ± 12.3	68.4 ± 12.4	68.8 ± 12.3
Non-ST-elevation myocardial infarction (NSTEMI)			
NSTEMI number (% of all ACS)	239 (52.6)	290 (56.5)	218 (41.0)
Age (mean ± SD)	69.6 ± 12.2	69.4 ± 11.9	64.6 ± 8.7
NSTEMI male (%)	163(68.2)	164 (56.6)	131(60.1)
Age (mean ± SD)	67.7 ± 11.7	66.3 ± 11.2	63.9 ± 9.4
NSTEMI female (%)	76 (31.8)	126 (43.4)	87(39.9)
Age (mean ± SD)	73.6 ± 12.3	73.4 ± 10.3	65.8 ± 7.5
Unstable angina pectoris (UAP)			
UAP number (% of all ACS)	103 (22.7)	25 (4.9)	42 (8.0)
Age (mean ± SD)	67.5 ± 11.3	64.4 ± 15.9	62,6 ± 14,8
UAP male (%)	66 (64.1)	18 (72.0)	26 (61.9)
Age (mean ± SD)	65.2 ± 10.4	61.5 ± 19.2	65.0 ± 17.3
UAP female (%)	37 (35.9)	7 (28)	16 (38.1)
Age (mean ± SD)	71.7 ± 11.8	72.0 ± 2.82	61.2 ± 15.0
Medical history (%)			
Acute coronary syndrome	21.8%	21.1%	21.3%
Percutaneous coronary			
intervention (PCI)	14.6%	17.1%	18.0%
Hypertension	80.7%	82.5%	83.1%
Diabetes mellitus	26.8%	33.5%	41.6%
Hyperlipidaemia	60.7%	69.8%	83.1%
Smoking	28.4%	30.4%	37.1%
Stroke	16.7%	13.5%	13.5%
Peripheral artery disease	31.0%	24.4%	21.3%
Percentage for follow-up (%)			
6 months	55%	54%	35%
12 months	73%	53%	43%
Lipid-lowering therapy at discharge (%)			
No statin	7%	1%	3%
Low-/moderate-intensity statin	1%	0	0
High-intensity statin	86%	96%	88%
Statin + ezetimibe	6%	3%	9%
PCSK9 inhibitors	0%	0%	0%

**Table 2 jcm-13-03398-t002:** The LDL-cholesterol levels and the 1.4 and 1.8 mmol/L LDL-cholesterol goal attainment rate based on the calculations with the Friedewald formula (LDL-F) and the Martin–Hopkins method (LDL-MH) in 2015, 2017 and in the COVID-19 year at the time of admission and 6 and 12 months after ACS.

	2015	2017	COVID-19 Year		
			Index Event	6 Months	12 Months
LDL-F (mmol/L, median [IQ-range])	1.99 (1.23–2.38)	2.42 (1.80–2.80)	3.20 (2.30–4.19)	1.64 (1.09–2.30)	1.60 (1.19–2.27)
LDL-MH (mmol/L, median [IQ-range])	2.04 (1.38–2.38)	2.55 (1.94–2.96)	3.32 (2.35–4.27)	1.92 (1.33–2.27)	1.73 (1.36–2.43)
The difference between LDL-F and LDL-MH	3%	5%	4%	−17%	−8%
Significance, *p*-value	NS	NS	NS	*p* = 0.044	*p* = 0.014
LDL-F 1.8 mmol/L attainment	37%	48%	32%	36%	LDL-F 1.8 mmol/L attainment
LDL-MH 1.8 mmol/L attainment	39%	46%	29%	32%	LDL-MH 1.8 mmol/L attainment
LDL-F 1.4 mmol/L attainment	16%	19%	22%	22%	LDL-F 1.4 mmol/L attainment
LDL-MH 1.4 mmol/L attainment	19%	20%	19%	26%	LDL-MH 1.4 mmol/L attainment

Note: *p*-value was calculated by the difference between the attainment rate of the LDL-C target calculated with LDL-F and LDL-MH. 2015 vs. COVID-19 year: LDL-F—NS; 2015 vs. COVID-19 year: LDL-MH—NS; 2017 vs. COVID-19 year: LDL-F—*p* =< 0.001; 2017 vs. COVID-19 year: LDL-MH—*p* =< 0.001.

## Data Availability

All data generated or analysed during the current study are available from the corresponding author on reasonable request.

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
