# Peer review of "The Evaluation of Lipid-Lowering Treatment in Patients with Acute Coronary Syndrome in a Hungarian Invasive Centre in 2015, 2017, and during the COVID-19 Pandemic—The Comparison of the Achieved LDL-Cholesterol Values Calculated with Friedewald and Martin–Hopkins Methods"

_jcm, 2024, doi:10.3390/jcm13123398_

Round 1
Reviewer 1 Report
Comments and Suggestions for Authors
Interesting study reporting the lipid lowering strategy in ACS patients from single Hungarian center in different references year, including the COVID-19 pandemic period. Of note, authors have reported the data also according to the Martin-Hopkins methods.
Results are consistent with poor goal achievement in LDL lowering strategy. This observational data focus on the large proportion of intervention still required in post ACS patients.
Some points could be improved:
A more characterized metabolic profile of patients should be reported, such as the proportion of hypertension, diabetes mellitus, obesity, MAFLD, glycemia, in a table.
Have any patients received PCSK9 inhibitors at discharge? Has someone during the follow-up period? In general, do authors have data about up titration of lipid lowering strategy at 6 months?
Report in the first row of table 1 and table 3 the number of patients with informative data about cholesterol measurement, as well as please specify and eventually split the data according to the year of enrollment and consider testing if any differences in the goal achievement exists.
Crucial interplay between lipid lowering therapy, platelet and inflammation exists, enhancing the cardiovascular risk: integrate the discussion (ie: PMID 36893777; 30150123; 24049520 )
Minor consideration:
- Did author have consideration about the incidence of statin-related myopathy and concomitant omega-3 supplements?
- Please shorten the introduction.
- Please report the figure 1 also according to the latest threshold of 1.40 mmol/L of LDL-C.
- Please use the same font for figure and text.
Reviewer 2 Report
Comments and Suggestions for Authors
Dear Authors
Thank you for your paper on the important issue of lipid-lowering treatment after ACS.
The main limitations result from the retrospective nature of the work, the size of the group, the method of follow-up and are discussed.
The work is carried out correctly and the results are well presented.
The only minor comments concern text editing. The work requires careful reading and corrections, for example line 107 "The We retrospectively collected data" or several references that are incomplete (no. 14, 20, 27)
Reviewer 3 Report
Comments and Suggestions for Authors
The paper covers important issue of cardiac care during COVID-19 pandemic, especially secondary prevention of ACS. The topic of the study is of high importance and significance, thus the pandemic was a terrible stress -test for health-care systems, including cardiovascular serves.
However, from my point of view, the paper requires extensive revision for better understanding and clearness.
1. The aim of the study should be clarified . The same about the title. The main idea as I got it is a rate of achievement of target lipids after ACS in COVID year in comparation with 2015 and 2017. Moreover, two formulas for LDL calculation were compared from this point of view. It seems to be two different studies.
One possible option is to calculate LDL levels via two formulas in 2015 and 2017 to present the complete data.
Other- to exclude Martin-Hopkins and present two formulas comparation as different paper.
2. Abstract. The result section has to be clarified according to corrections of result section in the main body of the paper (see above). Conclusions represent the mixture of the aims. However the study reveals that beside lower number of available samples (35% vs 54-55% at 6- months and 43% vs 73-53% at 12-months), the proportion of achieved target lipids is stable. That is important conclusion either.
3. Introduction section represents the scope of problem. The aim of the study has to be clarified (comparation with 2015 and 2017? Secondary aim for two formulas comparation&)
4. Materials and methods.
4.1 Patients.
1) Direct dates for 2015 and 2017 patients population inclusion should be clarified.
2) 6 and 12-months periods after discharge for data collection form databases should be mentioned
4.2 Determination of parameters. Sufficient
4.3 Calculation if LDL is well written
4.4 Statistic is essential.
5. Results.
1) Patients inclusion diagram should be added that would mention follow up.
2) It’s recommended to add Table 1 with patients characteristics for 2021, 2015, 2017 that would include age, sex, DM, STEMI/NSTEMI, as percentage for follow-up at 6 and 12 months with p-value between the groups. Moreover, current table 2 with lipid-lowering drugs prescription could be integrated into that table.
3) Current Table 1 is abundant. Actually, it shows the drugs work and the patients are complaint with the treatment. The table could be presented as supplementary material.
4) Tables 3 and 4 is a cornerstone of current publication. Figure 1 partly duplicates Table 4. The data requires decision how to present it according to first comment. If you have ability to calculate LDL-MN in 2015 and 2017 cohorts, I recommend to include parallel columns in Figure 1 for LDL-F and LDL-MN for 2015, 2017 and 2021 with p-values between the years and between the methods in one year.
6. Discussion is good. Figure 2 is of high importance.
7. Limitation section should added. One important limitation is unavailability of direct LDL-measurement, thus you can only make assumption that LDL-MD is more precised.
8. Conclusion have to be clarified
Comments on the Quality of English LanguageEditing is required
Round 2
Reviewer 3 Report
Comments and Suggestions for Authors
Paper is corrected. Could be recommended for the Journal
Author Response
Thank you for your positive response.